# Mothers’ Use of Regulatory Talk with Toddlers in Chile and the US: How Do Cultural Values and Children’s Gender Affect Mothers’ Regulatory Talk at 12 and 30 Months?

**DOI:** 10.3390/children8100874

**Published:** 2021-09-30

**Authors:** María Pía Santelices, Claire De Ann Vallotton, Chamarrita Farkas, Tzu-Fen Chang, Eduardo Franco, Ana María Gallardo

**Affiliations:** 1Psychology School, Pontificia Universidad Católica de Chile, Ave. Vicuna Mackenna 4860, Macul, Santiago 7820436, Chile; chfarkas@uc.cl (C.F.); efranco1@uc.cl (E.F.); amgallar@uc.cl (A.M.G.); 2Human Development and Family Studies, Michigan State University, East Lansing, MI 48502, USA; vallotto@msu.edu; 3Department of Child, Adolescent and Family Studies, California State University, Bakersfield, CA 93740, USA; tchang1@csub.edu

**Keywords:** mental state language, regulatory talk, regulatory attributes, gender socialization process

## Abstract

Studies have shown that Chilean and US infants differ in their levels of self-regulation. One of the mechanisms of early socializing is the use of language, particularly mental state language. The current study seeks to deepen our knowledge of the ways in which mental state language is related to socialization processes in early childhood, including the ways both culture and children’s gender influence a mothers’ use of mental state talk. We used a quantitative and descriptive approach with 109 mothers and their children (64 Chilean and 45 US dyads), measured twice, at 12 and 30 months old. Mental state references related to regulation were coded during a story-sharing task, including positive (calm and patient) and negative (messy and impatient) references to regulating behavior. Chilean mothers generally showed more regulatory references than US mothers, especially if the children were at a younger age (12 month). Frequencies of regulatory references increased in US mothers at 30 months but were still less than in Chilean mothers. At the 12-month measuring point, Chilean mothers showed more negative regulatory attributes than positive regulatory attributes. Finally, US mothers mainly used references to secondary emotions (e.g., pride) and positive regulatory attributes (being obedient, mature and patient) at both ages.

## 1. Introduction

The present study proposes a deeper understanding of the different types of mental state talk used by mothers during dyadic interaction with their children. Considering the different mental state talk types (emotions, cognitions, desires, perceptions, etc.) proposed for different studies [1,2,3,4,5], this paper seeks to focus on the mental state references that may control the behavior of the child by indicating the expected behavior (positive regulatory attribute language) or disapprove the undesired behavior (negative regulatory attribute language).

To achieve this, a regulatory index is proposed, which focuses on the analysis of the mother’s speech and the use of different types of mental state references (cognitions, emotions, attributes and moods) to influence a child’s behavior.

The following section presents a theoretical review of socialization processes through parenthood, the concept of mental state talk as a mechanism of socialization and a proposal to identify mental state references specifically aimed at promoting the regulation of the child’s behavior.

## 2. Background

### 2.1. Mentalization and Parental Mental State Talk

Mentalization refers to the ability to recognize thoughts, feelings, desires and intentions, both in oneself and in others, and to make accurate connections between observable behaviors and mental states [6,7,8]. Thus, a caregiver with adequate mentalization abilities will treat their child as a psychological agent with feelings, desires and intentions of their own, and will be able to consider their child’s desires and goals in a more accurate way [8,9,10].

Caregivers’ mentalization could also promote children’s understanding of the mental world—both their own and others. This enables the child to discover and understand his own internal experience and to become a mentalizing agent on his/her own [8,11,12]. When children have a caregiver who helps them to interpret their experiences, they gain security in their emotional bond and ability to self-regulate emotionally and behaviorally [13]. Thus, caregivers’ mentalization is related to children’s secure attachment [14,15,16], socio-emotional development [17,18] and theory of mind [19,20,21].

Mentalization develops and becomes evident through social interactions, with language favoring symbolic interaction. Therefore, language is not only one of the most relevant routes of socialization [22], but it is also through language that mental states gain meaning and can be communicated to others, enabling the interpretation of a certain behavior [11,13].

Thus, mentalization can be observed in verbal interactions—in this case, between an adult and an infant—through the presence of mental references in the adult’s discourse [23], with language being the mediator of the mentalization processes the adult transmits to the child through interaction [13]. Specifically, when adults put into words the internal states of the child, they convey an understanding of mental states by verbally interpreting their own behavior and that of others [11,13]. This encourages the child to label basic emotions [24] and enhances their own mentalizing ability [5,12,25], as well as their skills in empathy, perspective-taking and self-regulation.

Importantly, language is also a means to socialize [22] and control children’s behavior. The extent of this regulating language varies depending on parenting beliefs and styles, in accordance with the culture in which the family is immersed [26,27]. More specifically, regulatory speech—speech that acts to regulate children’s behavior—is used more frequently in collectivist cultures [28,29]. The current study examines whether mental state talk is used as part of this regulatory talk, and whether cultural differences in socialization are reflected in the way parents talk about the regulatory aspects of children’s internal states.

Mentalization is usually measured as the frequency or rate of parents’ use of speech that refers to the child’s emotions, cognitions and desires [1,4]; some studies also include perceptions [3], needs [2], attributes and/or causal talk [5].

Previous studies have found that the characteristics of maternal mental speech are influenced by multiple factors, including family socioeconomic level [30] and child characteristics, such as age [2,23,31] gender [31,32,33], temperament [27] and birth order [34]. Furthermore, mental state talk varies depending on context (e.g., play versus story reading) and the emotional nature of the event discussed [25,35].

Additionally, and more relevant in the context of the current research, Tamis-LeMonda and colleagues [29] explain that maternal mental speech also reflects cultural values. Specifically, the authors explain that Latino mothers could use more regulatory language to reflect the importance of children being “tranquilo”(quiet) and also the value of “respeto” (respect). Considering that mental state language contributes to the child-socialization process, it is expected that it will also contain regulatory language.

On the other hand, regarding parental mental speech, when comparing maternal and paternal mental state talk, maternal mental state talk shows a significant predictor in executive functions in pre-schoolers in contrast to the paternal interactions between fathers and toddlers [36].

However, not only is there a lack of research regarding mentalization in parenting within the Latin American context, but there is also a lack of cross-cultural studies that analyzes the use of mental state talk in different countries [1,37] with different values and parental attitudes. This is a very important issue, as Lecannelier [38] highlights, due to the fact that many South American countries—particularly Chile—are currently implementing prevention programs in early childhood. These programs could be developed from data provided by local research teams, which could help to understand how the use of mental state affects children’s socioemotional development within a culturally sensitive framework.

Research has documented that the ways in which parents display emotions, both verbally and nonverbally, influence children’s emotional behaviors and development [39,40,41]. Specifically, self-regulation, which is understood as a child’s capacity to regulate attention and behavior [42], is of particular importance because has been shown that children who show higher levels of flexibility or self-regulation display significantly fewer behavior problems than children who are temperamentally rigid or unadaptable [43], and that it is linked to socialization processes and environmental factors [44,45]. Emotion socialization has been shown to be a precursor to self-regulation in children, especially when maternal support in response to child distress through mentalization strategies is presented [46].

Thus, considering that parental mentalization is paramount in children’s socioemotional development, it is relevant to know which elements of mentalization—hereinafter referred to as mental state talk—might be associated with cultural and gender differences and describe the way it could shape children’s self-regulation skills.

### 2.2. Cultural Values and Child Socialization Process

Child socialization is the process through which children internalize beliefs, behaviors and feelings that are expected according to their position within their social group [47,48]. This learning is developed through the interactions with primary caregivers and others care figures that interact consistently with the child through daily activities.

The process of socialization is not seen as a passive internalization of certain structures external to the child’s identity. Instead, it is seen as a gradual process emerging from everyday interactions in a dynamic and continuous negotiation of activities and roles, where both the child and the adult are active players [47]. One of the main contributors to the evolution of this idea is Rogoff [49], who extends the approaches of Vigotsky [50] and raises the concept of “guided participation”, referring to the different ways in which children participate in their socialization process scaffolded by their caregivers. Children not only receive guidance from parents, but they also seek and demand this guidance in order to find solutions to everyday problems.

However, it is not possible to understand parenting without considering the contextual and cultural values in which it is immersed [51,52]. The general parameters of daily interrelation are established by cultural meanings, and parents reinforce cultural practices, depending on what is accepted or not within a culture [53]. This implies that each culture promotes its own forms of learning, aimed to achieve a variety of skills and values that define the culture’s goal of maturity [49,54]. Therefore, it is understood that parenting is positioned within the social structure of a given culture and is composed by a set of expectations and socially constructed tasks [55].

Roer-Stier [56] highlights that all cultures have an implicit definition of an “adapted” or “competent” person. This image serves to guide and organize the upbringing of children, with the objective that these children will fulfil the expectations of their culture and society. In this regard, these images are passed on from one generation to the other, and these patterns are stable and determine “successful parenthood” in each culture [57].

Another way to understand differences in parental practices could be related to the power distance (PD) hypothesis of Hofstede, Hofstede and Minkov [58]. This theory describes the extent to which members of a culture accept and expect power to be distributed unequally. This concept has been evaluated in different countries. Variation in the level of PD influences the way in which power is distributed in organizations and institutions, such as the family. According to this concept, countries have different needs in controlling their children’s behavior in order to meet cultural values and customs. They also differ in controlling emotional expression depending on social expectations [59]. On the one hand, individualistic cultures—represented by US and western cultures—have less PD, and encourage children to develop themselves, promoting autonomy and self-reliance in children [60]. In individualistic cultures, children are allowed and even encouraged to express their emotions [61], and caregivers tend to use a more authoritative parenting style, supervising child behavior to keep them safe while exploring, rather than using coercive control over children’s behavior [60]. On the other hand, collectivistic cultures—represented by Asian and Latin-American cultures—emphasize the interest of the community and interdependence as a main value. They inhibit the expression of individual needs over the needs of the group and establish a greater power distance [58]. To achieve these cultural expectations, caregivers use more authoritarian parenting styles as well as coercive control over children’s behavior [60]. 

Evidence for these theories is provided by studies in different cultural settings. For example, Cabrera, Shannon, West and Brooks-Gunn [62] report a greater use of the authoritarian parenting style and behavioral control in some Latin-American countries. In these cultures, authoritarian parenting is associated with positive child outcomes. Specifically, several studies have shown that maternal physical control is related to secure attachment in some Latin-American countries [63,64]. These findings suggest that authoritarian parenting, associated with behavioral control and greater PD, should be related to positive socialization in Latin-American countries, whereas in the US, it is associated with harsh parenting and poor child outcomes [65,66].

In the case of Chile, few studies have focused on parenting styles [67], which is one gap the current study will address. However, some reports [68] explain that, in general terms, Chilean parents tend to have a more paternalistic role in child-rearing practices, and behavioral control is paramount in their parenting style. This is coherent with a high level of PD in Chile, which has been considered a collectivistic country [58,69,70].

Nevertheless, although the current research addresses parenting with a dichotomist comprehension of culture (collectivistic vs. individualistic), it is important to highlight the fact that cultures between and within different countries are not homogeneous. In terms of parenting, parents from all backgrounds interact with their infants showing culture-specific patterns, and also showing different emphasis on communication [29].

Chile is an interesting case among Latin-American cultures. According to Ray and Peterson [67], the rapid development of the economy and changes in political and social systems in recent decades have led this country to move toward an individualistic pole in a short period of time. The authors explain that this could lead to changes in socialization processes, and possibly a decrease in authoritarian values. These broad social changes may explain the co-existence of both individualistic and collectivistic values within Chilean society and parenting [67]. These cultural shifts in parenting-related values reinforce the need to study parenting in a sociocultural context and the mechanism through which culture regulates the parent–child relationship.

However, it is relevant to emphasize the idea that cultural values shape children’s socialization processes in different and specific ways. One of these specific clusters is related to expectations in terms of gender roles and the gender socialization process. This topic is addressed below.

### 2.3. Cultural Values and Gender Socialization

It is relevant to consider gender-related differences in child-rearing practices in different cultures. Societies attribute different values and goals to each gender, which influences parents’ behavior towards the child from birth [48]. For example, some have studies found that both fathers and mothers tend to be more cognitively demanding when talking to their sons compared to their daughters [23,71,72], and to communicate concepts related to emotionality and fragility when talking to girls [72,73]. These communication patterns model behavior and establish limits that are internalized, becoming part of children’s gender identity [72,74]. Therefore, it is understood that the child socialization process varies between boys and girls from the very beginning of life in every culture.

Although previous theoretical background could eventually serve to hypothesize that Chilean families have different socialization gender expectations towards their children, there is no information about the use of maternal mentalization towards girls and boys.

Summarizing the previous ideas, the general and gender socialization processes vary from one culture to another. Although there might be various mechanisms underlying socialization processes [29], this study is focused on parental language.

As Schieffelin and Ochs [75] argue, “language is a major source of information for children learning the ways and world views of their culture”. The same authors explain that the primary concern of caregivers is to ensure that their children are able to display and understand appropriate behavior to social situations, language in everyday verbal exchanges being the best way to acquire sociocultural knowledge.

The current research questions concern a specific type of verbal interaction called mentalization or the use of mental state references, and its use in different cultures. This research elaborates on previous studies describing differences in the use of mental state references between Chile and the US, its impact on children’s outcomes [61] and the differences in the use of this type of language between girls and boys [33].

### 2.4. The Current Study: Regulatory Talk

The concept of regulatory talk has been used before at a representative level within the learning context of elementary and university students. Torres, Whitebread and McLellan [76] define this construct as the “set of communicative patterns engaged in by teachers to regulate children’s thinking and activity processes for learning” (p. 95). Other studies, even when not using the concept of regulatory talk, evaluate the relation between regulatory speech used by teachers and self-regulation in students [77,78]. Here, regulatory talk is linked to the mentalization process as a way of regulating and explaining mental states to others through conversation.

The idea of this paper has also been influenced by a previous study conducted by the same research team, which found that Chilean children show higher levels of self-regulation than US children at 12 months of age [61]. In the same sample, small cultural differences in mothers’ use of mental state references with infant boys and girls were identified, using a simple coding system with five categories of mental state references [33]. Specifically, US mothers used more causal talks with girls than with boys, whereas Chilean mothers spoke more about physical states with boys than with girls. The current study follows this line of research by analyzing mothers’ mental references with girls and boys at 12 and 30 months old, applying a more nuanced analysis of the mental references used by mothers in each country. The purpose of the current study is to focus on the use of regulatory talk—talk about internal states that may control the behavior of the child by indicating the expected behavior (positive regulatory attribute language) or disapprove the undesired behavior (negative regulatory attribute language).

To operationalize the concept of regulatory talk, a regulation index is proposed, with scores derived from mental content references that seek to guide the child’s behavior through the following: (1) cognitions, specifically, attention processes; (2) emotions, secondary emotions which allude to regulatory processes; (3) positive regulatory attributes, which aim to signal appropriate behavior; (4) negative regulatory attributes, which aim to signal inappropriate behavior; and (5) mood, temporary states that allude to regulatory processes. In the next section, examples of each category will be shown.

Based on the findings described above, three main hypotheses are derived:

(a) Chilean mothers use more regulatory elements in their speech than US mothers at both measuring points;

(b) More regulatory elements of speech are expected in both cultures at 30 months than at 12 months;

(c) Chilean mothers show greater gender differences in the use of mental language than mothers in the US across both ages.

Due to the low number of mental state references through a mother’s story telling task reported in a previous analysis with the same sample, it is expected that the number of words that appear under the concept of the regulation index will be sparse. Nevertheless, we consider it paramount to deepen understanding of child development from a gender perspective due to recent cultural changes. Previous studies regarding gender socialization have almost all been carried out in European or American populations. Thus, a comparison of gender socialization in developmental trajectories, comparing a Latin-American country (Chile) and the US, is novel in this line of research.

## 3. Method

### 3.1. Design

A nonexperimental, descriptive, longitudinal and comparative study was conducted describing mothers’ use of mental state talk in the interaction with their children, specifically considering their references to regulatory talk. Mothers were assessed at two stages: when their children were aged between 10 and 15 months and when they were aged between 28 and 33 months. Mothers’ regulatory talk was compared considering their country (Chile or the US) and children’s gender.

### 3.2. Participants

The present sample was drawn from a cross-national and longitudinal project that studied the relations between young children’s socioemotional development and maternal competences in Chile and the US. Participants were recruited from the childcare centers their children attended. These centers were chosen randomly, including both publicly and privately funded centers. The sample included 145 mothers and their infants (86 Chilean and 59 US dyads). In the Chilean group, the children were aged M = 12.01 months (SD = 1.39, range 10–15), 43.0% were girls and 57.0% were boys. The ages of the Chilean mothers ranged from 18 to 44 years, with M = 28.16 years (SD = 6.57). Of these, 18.6% did not finish high school, and from the remaining 81.4%, 16.3% had a technical education, 23.3% had a bachelor’s degree and 9.3% completed postgraduate studies. In the US sample, children were aged M = 12.07 months (SD = 1.48, range 10–15), 54.2% were girls and 45.8% were boys. The ages of US mothers ranged from 19 to 48 years, with M = 31.95 years (SD = 5.94). Of these, 98.3% completed high school, 8.5% had a technical education, 32.2% had a bachelor’s degree and 44.1% completed postgraduate studies.

This sample represents 82.9% of the original sample. Cases in which regulatory talk was not assessed at the beginning of the study, in which the adult assessed was not the mother and the mothers were younger than 18 years old, were excluded. Comparisons between these two final samples showed no differences in children’s age and gender, but the US mothers were significantly older—t = −3.54, *p* < 0.001—and had a higher educational level— X^2^ = 34.43, *p* < 0.001—in comparison with the Chilean mothers.

### 3.3. Assessments

*Sociodemographic variables.* To characterize both samples, a sociodemographic questionnaire was applied. This questionnaire was developed by the research team and included information about the child and parents, and contextual factors.

*Regulatory talk.* To assess this variable, the instrument chosen was the Evaluation of the Mentalization of the Significant Caregivers [1], which assesses adults’ mental state talk during storytelling interactions with the child. The adult receives two open-ended story-stems to tell their children. For example, “Andy/Mary is playing with the house keys, and he/she approaches the door. He/she tries to insert the keys, once and again, but he/she could not!”. Then, the adult is asked to develop the middle and end of the story based on each story-stem prompt. The instrument has two forms: Form A (children between 0–23 months) and Form B (children between 24 and 48 months). These forms vary based on story-stems that are more appropriate for each age, and the material provided to develop the stories (puppets in Form A, and pictures in Form B). The story-stems were developed in Chile and then were translated into English. Each story is based on a typical personal or interpersonal problem of young children, and the gender of the protagonist is altered to match the gender of the child subject. The whole situation is filmed, and then the adult’s speech is transcribed and coded. The instrument initially considers an event-based and mutually exclusive coding system of different categories of mental state talk (desires, cognitions, emotions, psychological attributes, mood states and physiological states) and categories of references that support the mental processes (causal talk, factual talk, links with the child’s life and physical expressions). Some of these categories include subcategories, and some of them were selected for this study.

In each country, the transcripts were coded by two independent coders who spoke the primary language of the participant (English or Spanish) fluently. Coders were trained by the developer of the coding system, and before the formal coding, the coders in each country practiced coding the same set of transcripts from the pilot studies in both Chile and the US. The transcripts from the other country were translated into the coders’ primary language. Independent coding started after a minimum of 80% cross-coder agreement was achieved for each category. For the Chilean sample, intercoder reliability was randomly assessed for 30% of videos, with correlations between two independent coders, ranging from 0.83 to 0.98 (M = 0.92). For the US sample, intercoder reliability was randomly assessed for 26% of the videos. Correlations between the two independent coders ranged from 0.74 to 0.99 (M = 0.87). Cohen’s kappa ranged from 0.55 to 0.70 (M = 0.63), and 0.50 to 0.94 (M = 0.78), respectively, for the Chilean and US samples, which represents a moderate to substantial agreement [79]. The validity of the coding system has been proved in previous studies where primary caregivers’ sensitivity was associated with socioemotional skills [1]. These analyses were obtained from the results of the longitudinal study within which the present study is framed.

For the present analysis, five types of references were included, considering their relation with regulatory talk: (a) attentional processes, which are cognitive processes, focused on the regulation (or not) of attention (e.g., pay attention to); (b) secondary emotions, which consider references to social emotions that encourage the control of behaviors for a particular goal (e.g., feel proud) or the emotional expression of emotions related to dysregulated behaviors (e.g., guilt or shame); (c) positive regulatory attributes (e.g., be obedient); (d) negative regulatory attributes (e.g., be disobedient); and (e) mood states that could be related to the expressions of regulated or dysregulated behaviors (e.g., feel bored) (see Table 1 for details). First, the frequency of mentions was obtained for each type of reference for both stories, then these frequencies were combined to measure the proportion of time engaged in negative attributes versus positive attributes for regulatory talk. Finally, to control the high variability in the number of words (range 45–2105 at 12 months, and 43–1366 at 30 months) and mothers’ verbosity, we calculated a proportion of regulatory talk over the total number of words in the mothers’ talk.

### 3.4. Procedure

Data for this study were obtained from a longitudinal study of socioemotional development from 12 to 30 months. Initially, different nurseries were randomly contacted and invited to participate in the study. Mothers (but also fathers and educational staff) whose children met the age requirements were informed about the study, and those who agreed to participate voluntarily signed an informed consent form. The assessments took place in the nurseries. Initially, the mothers completed the sociodemographic questionnaire and then were assessed with their children. The assessments were repeated when children were aged around 30 months of age.

### 3.5. Data Analysis

A mixed-method approach was taken, qualitative data were transcribed, and coded and statistical analyses were conducted with SPSS^®^ 19.0 for Microsoft Windows^®^. Initially, descriptive statistics were obtained for the characterization of the mothers’ regulatory talk for both countries, and both assessments of time (12 and 30 months). Then, analyses were conducted to compare mothers’ regulatory talk at 12 and 30 months considering their country (Chile or the US) and child’s gender (male or female), using the independent samples t-test. These analyses were repeated with ANCOVA, controlling for mothers’ age and education. Next, the paired samples *t*-test was used to compare the regulatory talk across time (12 or 30 months) for each country, and finally, comparative analyses were repeated with ANCOVA, controlling for mothers’ age and education, and analyzing the main effects and interactions for the time, country and child’s gender.

## 4. Results

### 4.1. Regulatory Talk in Mothers’ Speech When Children Are Aged 12 Months

When children were aged 12 months, the mothers’ proportion of regulatory talk was between 0 and 2.25 (M = 0.11, SD = 0.30) (see Table 2 for details considering country and gender). Comparative analyses showed significant differences between country—t = 3.97, *p* < 0.001—with a higher frequency of regulatory talk in Chilean mothers. Additionally, the analyses showed differences when the children’s genders were compared—t = 2.34, *p* = 0.021—indicating that mothers use more regulatory talk with boys.

Then, analyses were repeated with ANCOVA, controlling for mothers’ age and education, testing country, children’s gender, and the interaction among them. The model was significant—F (5, 139) = 3.66, *p* = 0.004, ηp^2^ = 0.116—and after including all the variables, the only variable that remained significant was country—F (1, 144) = 6.96, *p* = 0.009, η_p_^2^ = 0.048 (see Table 3 and Figure 1).

### 4.2. Regulatory Talk in Mothers’ Speech When Children Are Aged 30 Months

When children were aged 30 months, the mothers’ proportion of regulatory talk was between 0 and 1.41 (M = 0.13, SD = 0.27) (see Table 2 for details considering country and gender). Comparative analyses using a t-test first, and ANCOVA later, showed no differences between mothers’ regulatory talk and country or children’s gender at this age (see Figure 2).

### 4.3. Differences in Mothers’ Regulatory Talk between 12 and 30 Months

Analyses conducted with a paired samples t-test showed a significant increase in US mothers’ regulatory talk between 12 and 30 months—t = −2.31; *p* = 0.025—while no differences were observed for Chilean mothers or considering children’s gender.

Then, we conducted an ANCOVA analysis, controlling for mothers’ age and education, testing time (12 months, 30 months), country, children’s gender, and the interaction among them (see Table 4). The model was significant—F (9, 256) = 2.35, *p* = 0.015, ηp^2^ = 0.079—and after including all the variables, there was a significant interaction between time and country—F (1, 255) = 4.20, *p* = 0.042, ηp^2^ = 0.017. There was an increase in regulatory talk between 12 and 30 months in US mothers, while a decrease was observed in Chilean mothers between these ages (see Figure 3). Finally, children’s gender obtained a significant main effect—F (1, 255) = 4.16, *p* = 0.043, ηp^2^ = 0.017—indicating that when the scores for both times are considered, mothers use a higher frequency of regulatory talk with boys (M = 0.15 for boys; M = 0.07 for girls).

## 5. Discussion

Parents’ use of mental references in interaction with their children has been related to the development of the child’s own capacity to mentalize [5,15,25], promoting the later development of cognitive and socio-affective skills in the infant. As previous research indicated differences in regulation skills between Chilean and US children, in this study, we specifically evaluated the mental references aiming to control the children’s behavior included within the definition of the regulatory index. In this regard, we expected to find a greater number of regulatory references as the child grows and acquires a greater ability to regulate his/her behavior. Additionally, differences between the Chilean and US cultures were expected from the cultural parameters within which parenthood is framed and considering the results which show higher levels of self-regulation in Chilean children than US children [61].

To evaluate this variable, the storytelling context was chosen as it is considered as a task that promotes the use of mental states [80] and is an optimal activity to elicit references to mental states in adult speech during interactions with a child [81]. Studies have shown that the child’s age is a determining factor, finding an increase in mental references as the child grows [5,23,31]; however, this study showed different results, which are addressed in the following section.

Regarding the first hypothesis concerning differences between the two countries, we observed a significantly greater use of regulatory talk in the Chilean mothers than the US mothers at the first wave (at 12 months old). When the variables were measured separately, both at W1, the Chilean and the US mothers mainly used regulatory talk with boys rather than girls. Nevertheless, in another comparison, controlling for the mothers’ age and education, since these variables were not similar between both groups, it shows a significant difference between countries, with Chilean mothers scoring higher in regulatory talk. This could also be due to the fact that Chilean mothers belong to a more collectivistic culture and hence tend to be more authoritarian than mothers belonging to a more individualistic culture, as is the case with US mothers.

As mentioned before, at wave 2, there was a difference in the obtained results regarding the results of other studies that showed an increase in mental references according to the child’s age. In this study—as shown in Figure 2—the data analysis indicated that at 30 months, there was an increase in the use of regulatory talk in US mothers, unlike Chilean mothers, who presented a decrease in the use of this variable with their children between the ages of 12 and 30 months, even though this difference is not statistically significant. Hofstede and colleagues’ [58] hypothesis on power distance indicated that caregivers’ behaviors to control children’s behavior will be more culturally accepted in collective cultures. However, in our case, Chilean mothers used more regulatory elements than the US sample only at 12 months, which is a method to regulate children’s behavior, but afterwards, the US mothers showed an important increase in this variable, surpassing Chilean results in W2 and showing that collective cultures are not the only ones who accept and use regulatory elements. Additionally, an important aspect to analyze is the greater use of negative regulatory attributes (being naughty, messy, impatient, lazy, restless and hyperactive) in Chilean mothers at 12 months compared to the use of more positive regulatory attributes (being obedient, mature or “big” and patient) when the child was older by US mothers.

Regarding the second hypothesis (i.e., more regulatory references were expected at 30 months than at 12 months), Table 2 and Table 3 show that this is true for the US sample but not the Chilean sample, corroborated by χ2 analyses. The US mothers used very few regulatory references in the first wave while increasing their use in the second wave. This is consistent with the stage of development of their son or daughter. On the other hand, in the case of the Chilean mothers, it can be observed that there was even a slight decrease in regulatory talk in W2 in comparison to W1. This high use of regulatory references at an early age can be related to a need to control, as highlighted in the theoretical review [58]. However, this should be studied using longitudinal research to observe how regulatory talk changes across a wider child development span. This is relevant, mainly due to the negative correlation found between a style of restrictive control in the mother and self-regulation in different cultures during early childhood [30] and school-age children [82]

Considering the third hypothesis, proposing that Chilean mothers would show 68 more gender differences in the use of mental language than mothers in the US at both waves, our results did not support this idea. Contrary to our hypothesis, the findings show that, in both cultures, mothers show a greater number of regulatory references when talking to boys. However, this was significantly different only in W1 in contrast with W2. When analyzing changes between 12 and 30 months, looking at the variables separately, we can only see a significant increase in the use of regulatory talk in the US mothers. However, when we analyzed all the variables together—two waves, gender and country—and controlling for mothers’ age and education, we obtained two main results. First, there is a significant interaction between time and country, where the US mothers increase the use of regulatory talk through W2, while Chilean mothers start to score higher at 12 months and then reduce their use of regulatory talk with their children at 30 months. This could explain the significant difference shown at W1, as well as the disappearance of this difference at W2. Second, when considering both measures, the child’s gender appears to be significantly correlated with a major use of regulatory talk among boys, rather than girls.

This last outcome is an interesting result, which highlights how gender is an important element within socialization [72,74] and reinforces the results of previous studies that show differences in various aspects of how parents talk to their children depending on gender [31,32,33]. This could be explained by different levels of self-regulation [83,84] and variation in regulation temperament elements [85] in girls versus boys. Nevertheless, Mathew, Cameron and Morrison [86] highlight the impact of measurement in this type of study where observational methodologies could be susceptible to observer bias. Additionally, some results also propose the relevance of socialization in these differences; for example, the study carried out by McClelland, Acock and Morrison [87] evaluates the impact of kindergarten on children’s self-regulation abilities. Then, even when biological elements could result in boys needing more help to self-regulate, social elements should be further researched to understand this higher tendency to use regulation talk with boys in both countries.

In another line, when analyzing the types of categories used by mothers in each culture, we observed that Chilean mothers used more references to moods, cognitions and negative regulation attributes with boys and more cognitions with girls. In the case of US mothers, they mainly used references to cognitions, emotions and regulation attributes in the case of boys, while in the case of girls, they mainly used regulation attributes. Again, this is relevant not only due to the fact that Chilean mothers use a higher number of regulatory references at an early age, but also that, apparently, they use a higher percentage of negative regulatory attributes, while other mental references, such as talking about emotions and using positive regulatory attributes, were less common.

In general, it is important to highlight the existence of cultural differences and the relevance of investigation on this topic. It is important to gain a better understanding of how language is used as a means of socialization and how this specific type of language affects the development of social and regulation skills in children. Likewise, the differences in the use of regulatory references with boys and girls confirm previous results regarding differentiated socialization practices.

Finally, in relation to the limitations of the study, there are some aspects to discuss. First, it is important to consider that the length of the story that the adult creates in an incomplete story situation will affect the number of mental references he/she makes. In general, the lengths of the stories were related to an increase in the number and types of mental state references. Regarding this, even when we controlled for the total number of words in the stories in the ANCOVA analyses, the length of the story can reflect important elements of the mother’s use of mental state language. This limits the present study as shorter stories might hinder us from observing their use of mental state references. Second, the story designed for this study was aimed to elicit as many mental state references as the mother could utter. This form of measurement might limit the number of regulatory references, as can be seen by the small number of references reported. Nevertheless, the storytelling in this study represents a naturalistic interaction between the mother and child, showing that it is also probable that children in Chile and the United States listen to smaller amounts of regulatory references compared to other mental state references. However, future studies should consider designing instruments that elicit this particular form of mental references to confirm the results of this study. Third, it is important to consider the individual differences in relation to the tendency of each mother to use certain types of mental references. In this sense, even when the study compares mothers with boys versus mothers with girls, allowing us to evaluate general trends, future research could compare the same mother with children of different genders to reduce the incidence of individual differences. Fourth, the sample collected for this study was not very diverse in socioeconomic status (SES), showing higher levels of economic status in the United States sample than in the Chilean sample. There was also a difference in the educational level between countries, which could lead to shared variance between education and the country that should be studied in the future. Although all ANCOVA analyses were estimated with SES as a covariate, future studies should consider gathering more diverse samples in order to confirm that the results found in this study are due to cultural rather than economic differences. An important limitation of this study is the single inclusion of mothers as the main caregiver. Including both mothers and fathers in future studies would allow us to determine the contribution of each parental figure in the socialization of gender in different cultures, and also to compare the relationship of parents with children of the same gender versus those of different gender. Finally, only verbal communication was included, excluding nonverbal elements, which can be important to regulate children’s behavior, such as facial expressions, behavioral cues, physical affection and beating. Future investigation should include both verbal and nonverbal elements.

Regarding future research, it is expected that similar studies will be carried out in more cultures to understand intercultural similarities and differences in the practices of gender socialization of parents through language. Additionally, infuture studies it would be interesting to analyze whether experiencing motherhood for the first time would influence a mother’s rate of regulatory talk with their child. Similarly, it would be a great contribution to carry out studies with a longitudinal approach to assess changes in parental discourse throughout early childhood. Likewise, including parents or other caretakers, nonverbal elements and evaluating the same adult with different children, as indicated above, are possible lines of research from this study.

Finally, it is necessary to emphasize the relevance of regulatory talk, not only to the gradual acquisition of the child to self-regulate but also to the development of later social skills. The proposal of this new concept would also allow the evaluation of differences in the temperament and characteristics of each child, as well as in the socialization of gender in different cultures. As self-regulatory capacity is a relevant predictor of long-term development [88], a deeper understanding of how this capacity is acquired within early interactions would allow us to identify and promote positive parental practices.

## Figures and Tables

**Figure 1 children-08-00874-f001:**
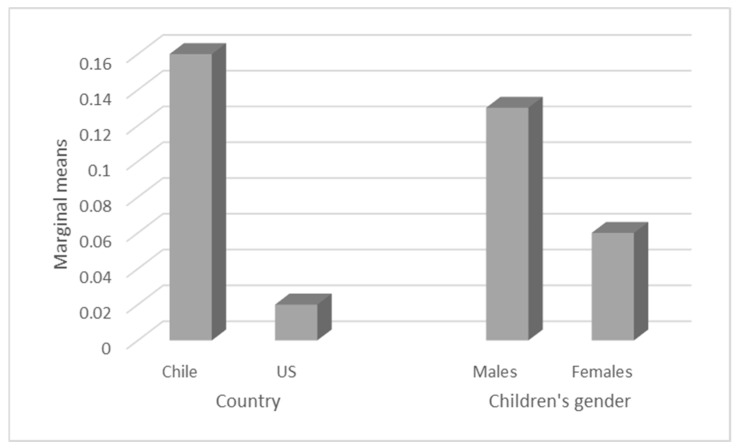
Distribution of the marginal means of regulatory talk at 12 months, controlled for country and children’s gender.

**Figure 2 children-08-00874-f002:**
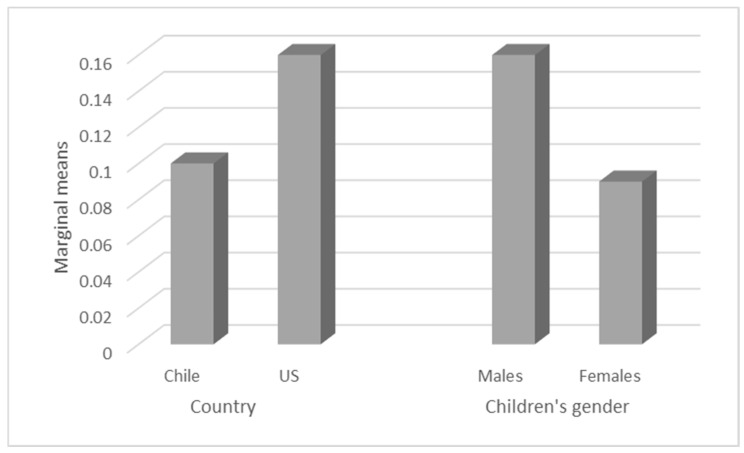
Distribution of marginal means of regulatory talk at 30 months, controlled for country and children’s gender.

**Figure 3 children-08-00874-f003:**
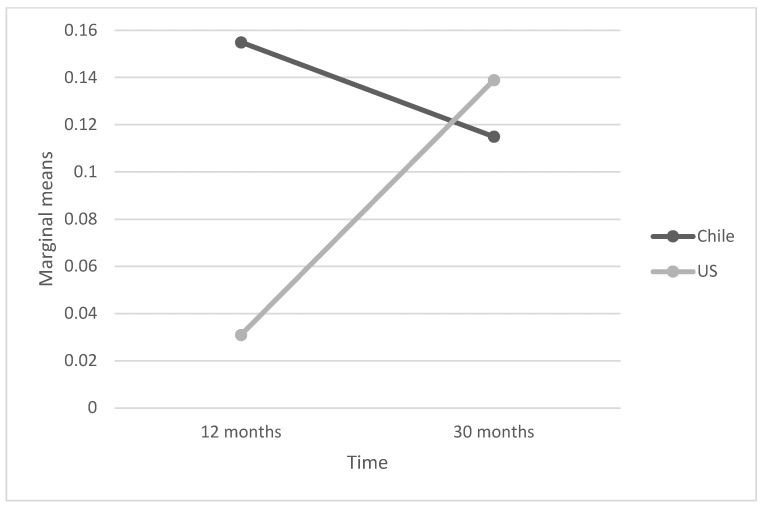
Distribution of marginal means of mothers’ regulatory talk considering time and country.

**Table 1 children-08-00874-t001:** Description of the categories considered in the study.

Category	Category or Subcategory Included	Examples
Cognitions	Attentional processes	Attend, concentrate, pay attention
Emotions	Secondary emotions	Proud, shame, guilt, embarrassed, worried
Psychological attributes	Positive regulatory attributes	Be obedient, patient, mature, careful
	Negative regulatory attributes	Be impatient, lazy, naughty, hyperactive, messy, disobedient
Mood states	Mood states	Feel calm, bored, entertained, motivated

**Table 2 children-08-00874-t002:** Descriptive statistics for mothers’ regulatory talk at 12 and 30 months, considering country and children’s gender.

	Chile				US			
	Male		Female		Male		Female	
	Min-Max	M (SD)	Min-Max	M (SD)	Min-Max	M (SD)	Min-Max	M (SD)
Regulatory talk at 12 m	0–2.25	0.24 (0.43)	0–1.05	0.09 (0.24)	0–0.26	0.03 (0.07)	0–0.24	0.01 (0.04)
Regulatory talk at 30 m	0–1.20	0.16 (0.30)	0–0.82	0.09 (0.21)	0–1.41	0.15 (0.32)	0–1.23	0.09 (0.26)

**Table 3 children-08-00874-t003:** Main and interaction effects (ANCOVA) of mothers’ country and children’s gender predicting mothers’ regulatory talk at 12 months (controlled for mothers’ age and education).

	F	*p*	df	η_p_^2^
Corrected Model	3.66	0.004	5	0.116
Intercept	0.11	0.745	1	0.001
Mothers’ age	1.24	0.268	1	0.009
Mothers’ education	0.75	0.389	1	0.005
Country	6.96	0.009	1	0.048
Children’s gender	2.25	0.136	1	0.016
Country x Children’s gender	1.22	0.271	1	0.009

N = 145.

**Table 4 children-08-00874-t004:** Main and interaction effects (ANCOVA) of time (12 or 30 m), mothers’ country and children’s gender predicting mothers’ regulatory talk (controlled for mothers’ age and education).

	F	*p*	df	η_p_^2^
Corrected Model	2.35	0.015	9	0.079
Intercept	4.27	0.040	1	0.017
Mothers’ age	0.11	0.746	1	0.000
Mothers’ education	1.69	0.194	1	0.007
Time	0.90	0.344	1	0.004
Country	1.46	0.229	1	0.006
Children’s gender	4.16	0.043	1	0.017
Time x country	4.20	0.042	1	0.017
Time x children’s gender	0.06	0.808	1	0.000
Country x children’s gender	0.71	0.399	1	0.003
Time x country x children’s gender	0.63	0.427	1	0.003

## Data Availability

Not applicable.

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
