# Peer review of "Mothers’ Use of Regulatory Talk with Toddlers in Chile and the US: How Do Cultural Values and Children’s Gender Affect Mothers’ Regulatory Talk at 12 and 30 Months?"

_children, 2021, doi:10.3390/children8100874_

Round 1
Reviewer 1 Report
Dear authors,
I have read the manuscript with interest, and value the effort. Saying this, I share with you quite a few comments and suggestions for your reflection and appreciation.
Please identify abbreviations used (e.g., SES; GEE).
It seems that a mixed-methods approach is addressed, and more than just “statistical analysis” was conducted, since it is referred that qualitative data were used, transcribed, and coded.
Detailed information about participants' selection would support the readers. What criteria are applied for the selection of the participants?
The option for a sample of mothers to the detriment of both parents is briefly referred to in the limitations section at the end. I suggest you add this dimension in the introduction.
Chile “has been considered a collectivist country”, but does it still being considered as so? A recent reference is needed to complement Hofstede et al. (1991) findings considering that twenty years have passed. Later this reference, a possible co-existence of both individualistic and collectivistic values is referred to, should this thesis be used for the discussion part, specifically in the first hypothesis’ discussion?
Are participants experiencing motherhood for the first time? This dimension appears to be interesting to explore.
Could a child’s characteristics affect mothers’ regulatory talk competencies and attitudes?
Is the “culture” dimension sufficiently explored and theorized to support the “culture effect” on mothers’ regulatory talk comparing Chile and the US? Should the title be reviewed in place or country concepts direction instead of “culture”? Keeping culture, perhaps “cultural values” could be more appropriate? And the research could include the assessment of the main values of each participant and then related them with the regulatory talk?
Saying this, the title, could it be reviewed according to the present comments and shorter?
A final reflection, some limitations of the study have been solved, namely by using triangulation of the research instruments, right?
All the best.
Author Response
|
Reviewer 1 |
Response |
|
Please identify abbreviations used (e.g., SES; GEE). |
Addressed.
SES: Social Economic Status GEE: generalized estimating equation |
|
It seems that a mixed-methods approach is addressed, and more than just “statistical analysis” was conducted, since it is referred that qualitative data were used, transcribed, and coded. |
Addressed. |
|
Detailed information about participants' selection would support the readers. What criteria are applied for the selection of the participants? |
Addressed.
Participants were recruited from the child care centers their children attended. These centers were chosen randomly, including both publicly and privately funded centers. |
|
The option for a sample of mothers to the detriment of both parents is briefly referred to in the limitations section at the end. I suggest you add this dimension in the introduction. |
Addressed.
In other hand, regarding parental mental speech, when comparing maternal and pa-ternal mental state talk, maternal mental state talk shows a significant predictor in execu-tive functions in pre-schoolers in difference with the paternal interactions between fathers and toddlers (Baptista, Osório, Costa Martins, Castiajo, Barreto, Mateus, Soares & Martins, 2017). |
|
Chile “has been considered a collectivist country”, but does it still being considered as so? A recent reference is needed to complement Hofstede et al. (1991) findings considering that twenty years have passed. Later this reference, a possible co-existence of both individualistic and collectivistic values is referred to, should this thesis be used for the discussion part, specifically in the first hypothesis’ discussion? |
Addressed.
Krassner, A. M., Gartstein, M. A., Park, C., Dragan, W. Ł., Lecannelier, F., & Putnam, S. P. (2017). East-West, Collectivist-Individualist: A Cross-Cultural Examination of Temperament in Toddlers from Chile, Poland, South Korea, and the U.S. The European journal of developmental psychology, 14(4), 449–464. https://doi.org/10.1080/17405629.2016.1236722 |
|
Are participants experiencing motherhood for the first time? This dimension appears to be interesting to explore. |
Addressed.
We proposed this in the future study section of the discussion. |
|
Could a child’s characteristics affect mothers’ regulatory talk competencies and attitudes? |
It is highly probable that the child’s characteristics could affect the mothers’ regulatory talk competencies and attitudes, nevertheless in this study we did not measure any child outcome. |
|
Is the “culture” dimension sufficiently explored and theorized to support the “culture effect” on mothers’ regulatory talk comparing Chile and the US? Should the title be reviewed in place or country concepts direction instead of “culture”? Keeping culture, perhaps “cultural values” could be more appropriate? And the research could include the assessment of the main values of each participant and then related them with the regulatory talk? Saying this, the title, could it be reviewed according to the present comments and shorter? |
Addressed.
Mothers’ use of regulatory talk with toddlers in Chile and the US: How does cultural values and child gender affect mother`s regulatory talk at 12 and 30 months? |
|
A final reflection, some limitations of the study have been solved, namely by using triangulation of the research instruments, right? |
Could have been solved by using another instrument to evaluate regulatory behavior in order to see if regulatory talk was associated to discipline behavior. |
Reviewer 2 Report
Summary: Mentalization, as part of inter-related parenting practices, is the ability of the parent to refer to one’s own and other’s internal states as well as recognize that one’s own mental state is different from another’s mental state. A parent’s ability to engage in mental state talk is associated with positive child outcomes over time including child regulation capacities. However, there are known differences in cultural values and experiences that often shape parenting behaviors. Because many countries are using early childhood interventions, it is important to understand similarities and differences in parenting practices, including mentalization, to inform culturally sensitive practice and optimize child outcomes. Dyads were assessed at two time points approximately 15 months apart. Caregiver’s regulatory talk (theoretically related to mentalization) was associated with education and country. Further, caregivers demonstrated change in regulatory talk over time and this change over time was different for US caregivers than Chilean caregivers.
Strengths:
- Cross cultural study of differences in parenting behaviors, specifically mentalization.
- Longitudinal design spanning 15 months.
- Utilization of rich qualitative data.
Major growth areas:
- The authors’ review of two very different topics is commendable – it is a difficult task. It is recommended that the theoretical explanation of mentalization occurs before talking about cultural differences in parenting. The authors articulate very clearly that there are cultural differences in parenting practices. However, in Line 151, the authors state ‘there is no information about the use of maternal mentalization towards girls and boys’. There is no context for this statement, and there is no operational definition of maternal mentalization before it is mentioned. This is important because parenting style, attachment, mentalization, and emotion socialization are all related, but different concepts – clearly explaining the theoretical background will set up for the next part of the introduction that some research indicates that there is some evidence for cultural differences in specific aspects of parenting.
- The choice to use self-regulation as an outcome needs to be explained a little more. There is a lot of evidence that emotion socialization (closely related to mentalization) is a precursor to self-regulation which is often a mediator of child outcomes; however, the authors do not explain this. While this is not a mechanisms paper, it is also important for the authors to explain the hypothesized mechanism for the association between mentalization and self-regulation – this will help to support the justification for using self-regulation.
- The discussion of regulatory talk (Line 254) also seems to be adjacently related to mentalization. Support for the current study will be strengthened by explicitly describing the link between mentalization and regulatory talk -or at least describe the hypothesized relationship between these two concepts.
- The Evaluation of Mentalization of the Significant Caregivers was used in the current study. Because it is a qualitative measure, it is important to give this measure some context by describing the reliability and validity of the measure (content validity – like it’s relation with other measures of mentalization, concurrent – it’s relation with other related measures), as well as how coders were trained and how they maintained fidelity to the coding system. Additionally, it is unclear why certain categories or sub-categories were included – specifically, does the system have a measurement for primary emotions and if so why were only secondary emotions included.
- In the descriptive statistics, the authors noted that US mothers had significantly higher education than Chilean mothers. Is there an interaction between education and country? It is possible that country is pulling shared variance between education and country.
- The types of regulatory talk seem to be different. Is it possible that a total score is not the most appropriate way to utilize this measure? Maybe proportion of time engaged in negative attributes versus positive attributes or attention to attentional processes versus emotions or mood states.
- Although the authors stated in Line 244 that self-regulation was an outcome in the current study, this paper did not discuss child outcomes related to the mentalization.
- Line 485 refers to an analysis that was not discussed in the results section. The authors stated that Chilean mothers had greater use of negative regulatory attributes than US mothers.
- The discussion refers to four hypotheses, but starting at Line 282 there were only three hypotheses reported.
Minor edits:
- Minor editing for readability is needed throughout the manuscript.
- Line 187 – 188 wording is confusing
- Line 195 wording
- Although kappa statistics of upwards of .94 are very good, it is inaccurate to say a kappa statistic of .50 represents substantial agreement. It is recommended that the range of moderate to substantial be included in this description.
Author Response
|
Reviewer 2 |
Responses |
|
The authors’ review of two very different topics is commendable – it is a difficult task. It is recommended that the theoretical explanation of mentalization occurs before talking about cultural differences in parenting. The authors articulate very clearly that there are cultural differences in parenting practices. However, in Line 151, the authors state ‘there is no information about the use of maternal mentalization towards girls and boys’. There is no context for this statement, and there is no operational definition of maternal mentalization before it is mentioned. This is important because parenting style, attachment, mentalization, and emotion socialization are all related, but different concepts – clearly explaining the theoretical background will set up for the next part of the introduction that some research indicates that there is some evidence for cultural differences in specific aspects of parenting.
|
Addressed. Mentalization has been changed before cultural differences so it should be clearer now on. |
|
The choice to use self-regulation as an outcome needs to be explained a little more. There is a lot of evidence that emotion socialization (closely related to mentalization) is a precursor to self-regulation which is often a mediator of child outcomes; however, the authors do not explain this. While this is not a mechanisms paper, it is also important for the authors to explain the hypothesized mechanism for the association between mentalization and self-regulation – this will help to support the justification for using self-regulation.
|
Addressed.
Emotion socialization has been shown to be a precursor to self-regulation in the child, especially when maternal support in response to child distress through mentalization strategies is presented (Cole, Dennis, Smith-Simon & Cohen, 2009). |
|
The discussion of regulatory talk (Line 254) also seems to be adjacently related to mentalization. Support for the current study will be strengthened by explicitly describing the link between mentalization and regulatory talk -or at least describe the hypothesized relationship between these two concepts.
|
Addressed. |
|
The Evaluation of Mentalization of the Significant Caregivers was used in the current study. Because it is a qualitative measure, it is important to give this measure some context by describing the reliability and validity of the measure (content validity – like it’s relation with other measures of mentalization, concurrent – it’s relation with other related measures), as well as how coders were trained and how they maintained fidelity to the coding system. |
We added information about coders being tained, but this was addressed in the text:
Coders were trained and before the formal coding, the coders in each country practiced coding the same set of transcripts from the pilot studies in both Chile and the US. The transcripts from the other country were translated into the coders’ primary language. In-dependent coding started after a minimum of 80% cross-coder agreement was achieved for each category. For the Chilean sample, inter-coder reliability was randomly assessed for 30% of videos, with correlations between two independents coders ranging from .83 to .98 (M = .92). For the US sample, inter-coder reliability was randomly assessed for 26% of the videos. Correlations between the two independent coders ranging from .74 to .99 (M = .87). Cohen’s kappa ranged from .55 to .70 (M = .63), and .50 to .94 (M =.78) respectively, for the Chilean and US samples, which represents a substantial agreement (Viera & Garrett, 2005). |
|
Additionally, it is unclear why certain categories or sub-categories were included – specifically, does the system have a measurement for primary emotions and if so why were only secondary emotions included.
|
The sub-categories mentioned in the study were include because of their relationship with regulatory talk.
In the case of secondary emotions, they were included because they “consider the references to social emotions that encourage the control of the behaviors for a particular goal (e.g., feel proud) or the emotional expression of emotions related with dysregulated behaviors (e.g., guilt, ashamed)”, as mentioned in the article. |
|
In the descriptive statistics, the authors noted that US mothers had significantly higher education than Chilean mothers. Is there an interaction between education and country? It is possible that country is pulling shared variance between education and country. |
Addressed in the discussion in the limitation section.
There was also a difference in the educational level between countries, which could lead to shared variance between education and country that should be studied ahead. |
|
The types of regulatory talk seem to be different. Is it possible that a total score is not the most appropriate way to utilize this measure? Maybe proportion of time engaged in negative attributes versus positive attributes or attention to attentional processes versus emotions or mood states.
|
Addressed.
Changed to proportion of time engaged in negative attributes versus positive attributes. |
|
Although the authors stated in Line 244 that self-regulation was an outcome in the current study, this paper did not discuss child outcomes related to the mentalization.
|
In the discussion, there are references for child self-regulation according to the cultures involved in this study. This paper was focused in measuring regulatory talk in mothers, but not child outcomes related to mentalization. Nevertheless, it is mentioned that in order to observe how regulatory talk influences in self-regulation outcomes across the development of a child, there should be used a longitudinal study instead. |
|
Line 485 refers to an analysis that was not discussed in the results section. The authors stated that Chilean mothers had greater use of negative regulatory attributes than US mothers.
|
In the result section, there were only exposed the cuantitative results, but not the cualitative obtained by the coders |
|
The discussion refers to four hypotheses, but starting at Line 282 there were only three hypotheses reported.
|
The discussion only refers to the main three hypotheses. Paragraph starting at line 548 does not state for a fourth hypothesis. In the limitation section there are exposed five points, but none of them refer to the hypotheses of the present study. |
|
Line 187 – 188 wording is confusing
|
Addressed.
but it is also through language that mental states gain meaning and can be communicated to others, transmitting ones’ interpretation concerning a certain behaviour |
|
Line 195 wording
|
Addressed.
they convey an understanding of mental states, by verbally interpreting their own behavior and that of others |
|
Although kappa statistics of upwards of .94 are very good, it is inaccurate to say a kappa statistic of .50 represents substantial agreement. It is recommended that the range of moderate to substantial be included in this description.
|
Addressed. Added in the description. |

Round 2
Reviewer 2 Report
- The manuscript was re-organized as suggested – flows much better.
- Lines 131 – 134. This sentence suggests that the current (Santelices et al.,) study will use child self-regulation. I suggest rewording to something like this: Specifically, self-regulation -which is understood as a child’s capacity to regulate attention and behavior- (cite) is of particular importance because research indicates that children who are higher in flexibility or self-regulation show significantly fewer behavior problems…
- The operational definition of regulatory talk was helpful.
- The authors did address some concerns about training. However, it is unclear if the coders were trained by a trained coder or the developer of the coding system.
- Please put in a sentence or two about validity of the coding system: specifically, previously published literature on 1) if the measure/coding system is highly correlated to other measures of mentalization and 2) if the measure/coding system is correlated with current or future functioning (like parental sensitivity, child social-emotional development, etc.).
- It is unclear which subcategories were considered as negative attributes versus positive attributes. (Line 384 – 387). Were all sub-categories considered positive or negative or were only the positive and negative considered. The proportion of time spent using either positive or negative attributes was not reported and there are no descriptive statistics reported using these variables (child age, country, education, child gender, ect.). There are no analyses reported in the results section suggesting that positive versus negative attributes were conducted (all analyses seem to be the total proportion of regulatory talk conducted).
- Line 502 – 505. Is this referring to future directions, a study that has already been conducted, or the current study? If these are results from the current study, they should be put in the results section before mentioning them in the discussion.
- Lines 550 – 558. If these are results from the current study, they should be put in the results section before mentioning them in the discussion.
- Line 570. If these are results from the current study, they should be put in the results section before mentioning them in the discussion. If the GEE model is different than an ANCOVA, it should also be mentioned in the data analysis section.
- Maybe a sub-heading under the results section labeled “exploratory analyses” or “follow-up analyses” should be added to accommodate the additional analyses that were not included in the current iteration of the results section. Also, a rationale should be given as to why the follow-up analyses were conducted.
Author Response
|
Reviewer 2 |
Comments |
|
Lines 131 – 134. This sentence suggests that the current (Santelices et al.,) study will use child self-regulation. I suggest rewording to something like this: Specifically, self-regulation -which is understood as a child’s capacity to regulate attention and behavior- (cite) is of particular importance because research indicates that children who are higher in flexibility or self-regulation show significantly fewer behavior problems…
|
Addressed. Changed. |
|
The authors did address some concerns about training. However, it is unclear if the coders were trained by a trained coder or the developer of the coding system.
|
Addressed. |
|
Please put in a sentence or two about validity of the coding system: specifically, previously published literature on 1) if the measure/coding system is highly correlated to other measures of mentalization and 2) if the measure/coding system is correlated with current or future functioning (like parental sensitivity, child social-emotional development, etc.).
|
Addressed.
Farkas, C., Vallotton, C. D., Strasser, K., Santelices, M. P., & Himmel, E. (2017). Socioemotional skills between 12 and 30 months of age on Chilean children: When do the competences of adults matter?. Infant behavior and development, 49, 192-203. |
|
It is unclear which subcategories were considered as negative attributes versus positive attributes. (Line 384 – 387). Were all sub-categories considered positive or negative or were only the positive and negative considered. The proportion of time spent using either positive or negative attributes was not reported and there are no descriptive statistics reported using these variables (child age, country, education, child gender, ect.). There are no analyses reported in the results section suggesting that positive versus negative attributes were conducted (all analyses seem to be the total proportion of regulatory talk conducted).
|
In table 1 is exposed that the positive regulatory attributes are those like: Being obedient, patient, mature, careful. And the negative regulatory attributes are those like: Being impatient, lazy, naughty, hyperactive, messy, disobedient.
There are only 2 subcategories considered as such, but the others subcategories are not considered positive or negative, because they respond to attentional processes, secondary emotions and mood states, which are not catalogued as positive or negative. That is why there is no analyses reported suggesting positive versus negative attributes. |
|
Line 502 – 505. Is this referring to future directions, a study that has already been conducted, or the current study? If these are results from the current study, they should be put in the results section before mentioning them in the discussion.
|
They are referring to future directions in further studies |
|
Lines 550 – 558. If these are results from the current study, they should be put in the results section before mentioning them in the discussion.
|
Lines 550 to 558 refers to future research. |
|
Line 570. If these are results from the current study, they should be put in the results section before mentioning them in the discussion. If the GEE model is different than an ANCOVA, it should also be mentioned in the data analysis section.
|
There was a mistake in the manuscript. We changed “GEE models” for “ANCOVA analyses”, because there were no GEE models in the results. Now it should be understood better. |
|
Maybe a sub-heading under the results section labeled “exploratory analyses” or “follow-up analyses” should be added to accommodate the additional analyses that were not included in the current iteration of the results section. Also, a rationale should be given as to why the follow-up analyses were conducted.
|
In this study it was decided only to report the ANCOVA, so there are no additional analysis. |
